# The Influence of Listening to Preferred versus Non-Preferred Music on Static and Dynamic Balance in Middle-Aged Women

**DOI:** 10.3390/healthcare11192681

**Published:** 2023-10-03

**Authors:** Fatma Ben Waer, Cristina Ioana Alexe, Dragoș Ioan Tohănean, Denis Čaušević, Dan Iulian Alexe, Sonia Sahli

**Affiliations:** 1Research Laboratory Education, Motricité, Sport et Santé, LR19JS01, High Institute of Sport and Physical Education of Sfax, University of Sfax, Sfax 3000, Tunisia; fatmaelwaer123@gmail.com (F.B.W.); sonia.sahli@isseps.usf.tn (S.S.); 2Department of Physical Education and Sports Performance, “Vasile Alecsandri”, University of Bacău, 600115 Bacău, Romania; 3Department of Motric Performance, “Transilvania” University of Brașov, 600115 Brașov, Romania; dragos.tohanean@unitbv.ro; 4Faculty of Sport and Physical Education, University of Sarajevo, 71000 Sarajevo, Bosnia and Herzegovina; denis.causevic@fasto.unsa.ba; 5Department of Physical and Occupational Therapy, “Vasile Alecsandri”, University of Bacău, 600115 Bacău, Romania

**Keywords:** self-selected music, postural balance, women

## Abstract

Although many women perform postural tasks while listening to music, no study has investigated whether preferred music has different effects than non-preferred music. Thus, this study aimed to explore the effects of listening to preferred versus non-preferred music on postural balance among middle-aged women. Twenty-four women aged between 50 and 55 years were recruited for this study. To assess their static balance, a stabilometric platform was used, recording the mean center of pressure velocity (CoPVm), whereas the timed up and go test (TUGT) was used to assess their dynamic balance. The results showed that listening to their preferred music significantly decreased their CoPVm values (in the firm-surface/eyes-open (EO) condition: (*p* < 0.05; 95% CI [−0.01, 2.17])). In contrast, when the women were listening to non-preferred music, their CoPVm values significantly (*p* < 0.05) increased compared to the no-music condition in all the postural conditions except for the firm-surface/EO condition. In conclusion, listening to music has unique effects on postural performance, and these effects depend on the genre of music. Listening to preferred music improved both static and dynamic balance in middle-aged women, whereas listening to non-preferred music negatively affected these performances, even in challenged postural conditions.

## 1. Introduction

The health of middle-aged women is a major public health concern worldwide [1]. Middle age is an important period because it involves unbearable difficulties, especially for women, such as worries about growing old, marital stress, widowhood, retirement, or family deaths. These women are often referred to as the “sandwich generation”, as they play a significant role in providing financial support and care for their children, grandchildren, and aging parents [2,3]. During this period, women often undergo many physical and psychological changes due to menopause [4]. Most of these changes alter health-related fitness, including strength, muscle endurance and mass, flexibility, and cardiovascular endurance [5,6]. Particularly, a significant decline in postural balance has been observed in women aged between 40 and 60 years [7].

These postural declines in middle-aged women worsen with menopause [8] and, consequently, affect their personal and functional autonomy linked to their individual capacity to carry out daily activities [9], leading to a high risk of falls [10]. It has been reported that falls are the third-leading cause of unintentional injury deaths among people aged between 45 and 64 years [11,12]. Moreover, fall-related injuries in elderly women are more severe, and their costs are estimated to be two to three times higher than those of fall injuries in older men [13]. These falls can seriously impact women’s quality of life and lead to high morbidity and mortality, as well as increased direct costs to health services [14]. Indeed, falls have dramatic consequences for the individual who suffers them, in terms of physical impairment as well as psychological and social problems. The combination of these three consequences causes an increased functional decline, ultimately leading to morbidity [15]. Quality of life is, therefore, affected by falls, whether traumatic or not [16].

The loss of postural balance can be induced by multiple factors, like the deterioration of several systems, such as the sensory, musculoskeletal, and neuromuscular systems, central nervous system (CNS) impairments, and perturbations in movement and orientation strategies [17]. Postural balance is a complex motor skill derived from the interaction between the visual, somatosensory, and vestibular systems [17], which is commonly considered the major input [18]. It has also been shown that auditory information is involved in the regulation of postural balance, both in healthy people and in those suffering from pathological conditions [19]. Phonoreceptors and the vestibular organ are, anatomically and functionally, mutually connected. The auditory input receives sound stimuli in the form of an air-density wave, which, in turn, is believed to influence postural regulation [10]. In this context, a previous study showed that when vestibular inputs are disrupted, the auditory system provides balance-related cues that reduce body sway by up to 41% [13]. 

It is important to note that, in daily life, most individuals tend to listen to music. Music has been widely used as a nursing intervention to bring multiple benefits in terms of physical and psychological well-being. Indeed, listening to music captures attention, raises spirits, triggers a range of emotions, boosts motivation, regulates mood, evokes memories, increases work output, heightens arousal, induces states of higher functioning, promotes motor coordination, and encourages rhythmic movements [20,21]. Researchers have claimed that the brain’s role as the main regulator of locomotion, neurovascular control, and sensory integration explains the connection between music tempo and physiological processes. More precisely, there could be a “pattern generator” underlying locomotor rhythmicity.

It acts as a specific auditory stimulus to elicit emotional and motor responses by activating a variety of brain area regions (the auditory nerve, brainstem, thalamus, and auditory cortex) [22]. In particular, a significantly straightened posture, stronger and more symmetric movements, and an increased awareness of the self and one’s environment have been reported while listening to music [23]. Considering this, many investigations have been conducted to determine the musical benefits (e.g., of relaxing music, the Bluebell Polka, and classical music) on postural balance in different populations: healthy subjects [22,24], young [25] and older adults [26], and patients [27,28]. However, controversial results have been found. Indeed, some of these studies reported a potentially positive effect on both static [22,24] and dynamic [27] postural balance, while others did not [25]. It is important to note that most people tend to listen to their preferred music in their daily lives. It was found that listening to preferred music induced pleasant feelings of energy, a positive mood, and a reduction in agitated behavior [29,30,31]. Further, listening to preferred music enhanced postural balance in visually impaired adolescents [32], whereas listening to non-preferred music resulted in a worsened cognitive performance. In contrast, Forti et al. (2010) found no significant effects of preferred music on postural balance in individuals with typical development [22]. However, a previous study also demonstrated that preferred music is an important mediator of the music’s ergogenic potential, whereby listening to preferred music improves both performance and psychological factors compared to listening to non-preferred music [33,34]. It, therefore, seems that the choice of music may play an important role in determining whether music acts as an ergogenic aid.

Despite the positive effects of self-selected music, to our knowledge, there are limited data concerning its effect on postural balance. Given that more attention may need to be paid to middle-aged women, as an overview of postural impairment in this group is potentially important to delaying problems later in life, our aim is to evaluate the effects of listening to preferred versus non-preferred music on postural balance in these women. We hypothesized that preferred music would enhance postural performance in these women and that these effects would be absent while the women were listening to non-preferred music.

## 2. Materials and Methods

### 2.1. Participants

According to Beck, G*power software (version 3.1.9.2; Kiel University, Kiel, Germany) was used to calculate the required sample size [35]. Values for power, correlations among repeated measures over the group, and the non-sphericity correction (ε) were set at 0.95, 0.5, and 1, respectively. While there is a lack of data regarding the effects of music on postural performance, an a priori power estimation was conducted using a large effect size of Cohen f (0.4) for an analysis of variance with one group and six measurements to aid in clinical interpretability of the results. This calculation led to a required sample size of at least 12 participants in order to minimize the risk of Type II statistical error. We recruited 24 participants (age: 52.5 (±2.7) years; height: 1.56 (±0.8) m; weight: 78.3 (±6.5) kg) who met the inclusion criteria from the general community to participate in our study. These inclusion criteria were healthy, aged between 50 and 55 years, post-menopausal for at least 4 years, with a mild risk for falling. All women were physically independent with no physical or mental illness, including any orthopedic, musculoskeletal, neurological, or respiratory dysfunctions or visual and/or vestibular disorders. Participants with grade III obesity, uncontrolled hypertension, and ingestion of medication that can affect postural balance outcomes were excluded.

The experimental protocol, with its risks and benefits, was clearly explained to all participants. Subsequently, all women gave their written informed consent prior to participating in this study. This study was conducted according to the Declaration of Helsinki and was fully approved by the Ethics Committee of the Vasile Alecsandri University of Bacău Romania.

### 2.2. Study Design

A randomized, counterbalanced crossover design with three conditions (preferred music, non-preferred music, and a no-music control) was adopted. Participants were welcomed to the laboratory in three sessions separated by at least 4 days. Upon arrival to the experimental room, participants were instructed to bring two pieces of music from their own playlists on the days of experiment. One piece should be their most preferred music and one the most non-preferred one, and the pieces should be both familiar and with lyrics. A certified music therapist has provided full details of the preferred and non-preferred music selected by the participants (see Appendix A). The genres of the most preferred type of music were classical (64%) and county (36%), whereas the most non-preferred type of music was rap (72%) and jazz (28%). Yet, these self-selected songs differed in tempo mode and energy. The first session was the familiarization session that was conducted 3 days before beginning the experimental protocol. During this session, all participants performed a short trial (about 10 s) for each task in the non-music condition without assessing any music effects to ensure that they were familiarized with the experimental protocol. The second and third sessions were the testing sessions in which we assessed the postural balance of all the participants ((static balance: in an upright bipedal stance during different sensory conditions), dynamic balance through performing the timed up and go test (TUGT)) in three auditory conditions (no music (absence of auditory stimulus) and preferred vs. non-preferred music (self-selected by participants)) by wearing headphones (Figure 1). These postural tests were performed in a randomized order to avoid the influence of learning or fatigue on outcomes by the same experimenters who were blinded to auditory conditions. 

The same smartphone and headset were used by each participant. The music volume was set as 10/15 with an average of 65 ± 5 decibels (dB) measured using an Android application (Sound Meter (ver. 1.6.5a)) in each test [25]. This protocol was carried out based on the Reporting Guidelines for Music-based Interventions [36].

### 2.3. Postural Balance Assessment

#### 2.3.1. Static Balance

Postural Balance was evaluated using a static stabilometric platform (posture Win©, Techno Concept^®^, Cereste, France; 40 Hz frequency, 12 bit A/D conversion) (Figure 2). It records the center of pressure (CoP) sways with three strain gauges. Participants were instructed to stand barefoot, as immobile as possible, on the force platform in an upright bipedal posture with their arms comfortably positioned downward at either side of the body. Their bare feet were separated by an angle of 30° while their heels were placed 5 cm apart. Postural measurements were collected in two vision conditions. In the eyes open (EO) condition, participants were instructed to keep their gaze horizontal in a visual target positioned 2 m away. 

Postural measurements were recorded in two vision conditions and two surface conditions. In the eye open (EO) condition, participants were asked to keep their gaze horizontal in a visual target positioned 2 m away, whereas in the eyes closed (EC) condition, the vision was eliminated by wearing a blindfold. For each of the eye conditions, participants were asked to maintain an upright bipedal posture on two surface conditions: firm surface (the rigid surface of the force platform) and foam surface (surface consisted of a foam block (466 mm length × 467 mm width × 134 mm height above ground) with a density of 21.3 kg/m^3^ and an elastic modulus of 20.900 N/m^2^ [37], mounted on the rigid surface of the force platform). Three trials were conducted for each experimental condition. According to the French Posturology Association norms, the duration of each trial is 30 s, with 30 s of rest between trials. All experiments were evaluated by the same experimenter who stayed near the participant for security without adducing any additional directions. The CoP mean velocity (CoP_Vm_) was selected as it is the most accurate form of sensory information used to stabilize posture during a quiet stance [38]. It corresponds to the sum of the CoP displacement scalars divided by the sampling time. The best postural balance is for the lower values of this CoP_Vm_ parameter [39].

The Romberg index (RI) scores were also calculated [40,41]. It evaluated the contribution of different sensory conditions (vision and auditory inputs) to maintaining posture control and the relevant ratio: (a) auditory manipulation with eyes open (Index 1; transition from eyes open to eyes open with auditory manipulation (e.g., listening to preferred or non-preferred music); EOAM/EO ratio); and (b) auditory manipulation with eyes closed (Index 2; transition of eyes closed to eyes closed with auditory manipulation (i.e., listening to preferred or non-preferred music); ECAM/EC ratio. The RI score by the relevant ratio standardized to unit value 1 and transformed to a percentage (e.g., Index 1 score on total displacement = ((Total displacement EOAM in the music condition/Total displacement EO in the non-music condition) − 1 × 100).

#### 2.3.2. Dynamic Balance

The timed up and go test (TUGT) was performed to assess dynamic balance in middle-aged women [42] (Figure 1). The TUGT is an internationally accepted functional dynamic test of balance, which has been well-proven to be reliable, valid, low-cost, and easy to apply [43,44]. Each participant was asked to sit with her back against the chair, her arms resting on the armrests of the chair, wearing her regular footwear. They were also instructed, at the word “Go”, to stand up and walk at a comfortable, safe pace to a line drawn on the floor 3 m away, turn around, return to the chair, and sit down again. The time taken from the command “Go” to when the participant was sitting with their back resting against the back of the chair was recorded, in seconds, using a stopwatch [45]. 

Each participant was instructed to sit with her back against the chair, arms resting on the armrests of the chair, wearing her regular footwear. At the word “Go”, they were also asked to stand up and walk at a comfortable and safe pace to a line drawn on the floor 3 m away, turn, return to the chair, and sit down again (Figure 1). The time taken from the command “Go” to when participants were sitting with their back resting against the back of the chair was recorded in seconds, using a stopwatch [45]. 

### 2.4. Statistical Analyses

The statistical analyses were performed using the software Statistica 12 (StatSoft, France). Values were expressed as means ± standard deviations (SD). The Shapiro–Wilk test reported that data were normally distributed. The Levene test was applied to verify the variance homogeneity. Once the homogeneity was verified, a three-way ANOVA with repeated measures (2 vision × 2 surfaces × 3 auditory conditions) was used to determine the effects of the auditory conditions (preferred music, non-preferred music, and a no-music control), vision (EO/EC) and surfaces (Firm surface and Foam surface) factors on the CoP_Vm_ values. When significant differences (*p* < 0.05) were observed, a post-hoc analysis was then performed with a Bonferroni significant difference test [46]. Additionally, the TUGT scores were analyzed using a one-way ANOVA with repeated measures (preferred music, non-preferred music, and a no-music control). The partial eta squared (η^2^_p_) was executed to calculate the effect sizes for the main and interaction effects (small effect: 0.01 < η^2^_p_ < 0.06; medium effect: 0.06 < η^2^_p_ < 0.14; and large effect: η^2^_p_ > 0.14), and the Cohen’s d for the pairwise differences (trivial: d < 0.2; small: 0.2 ≤ d < 0.5; moderate: 0.5 ≤ d < 0.8; large d ≤ 0.8) [47]. In addition, A 90% confidence interval (CI) for ηp^2^ and 95% CI for each comparison were performed [48]. 

## 3. Results

### 3.1. Static Balance

The three-way ANOVA showed a significant main effect of auditory conditions (F_(2,50)_ = 26.65, *p* < 0.0001, η^2^_p_ = 0.51), surface (F_(1,25)_ = 315.85, *p* < 0.001, η^2^_p_ = 0.92) and vision (F_(1,25)_ = 364.25, *p* < 0.0001, η^2^_p_ = 0.93) factors as well as a significant surface × auditory conditions (F_(2,50)_ = 7.82, *p* < 0.01, η^2^_p_ = 0.23), vision × auditory conditions (F_(2,50)_ = 8.11, *p* < 0.001, η^2^_p_ = 0.24) and surface × vision (F_(1,25)_ = 28.31, *p* < 0.001, η^2^_p_ = 0.53) interactions on the CoP_Vm_ values with large effect size (η^2^_p_ > 0.14). However, no significant surface × vision × auditory condition interaction was observed.

Concerning auditory condition factor, the post hoc results showed that the CoP_Vm_ values significantly decreased when listening to preferred music compared to the no-music [in the firm surface/EO condition: (*p* < 0.05; 95% CI [−0.01, 2.17]), and the non-preferred music (firm surface/EO: (*p* < 0.01; 95% CI [0.19, 2.39]); firm surface/EC: (*p* < 0.001; 95% CI [0.6, 2.8]); foam surface/EO: (*p* < 0.001; 95% CI [0.38, 2.57]); foam surface/EC: (*p* < 0.001; 95% CI [1.79, 3.99])] conditions (Table 1 and Figure 3). In contrast, when listening to non-preferred music, the CoP_Vm_ values significantly (*p* < 0.05) increased compared to the no-music condition in all the postural conditions [firm surface/EC: (*p* < 0.001; 95% CI [−2.67, −0.47]); foam surface/EO: (*p* < 0.001; 95% CI [−2.6, −0.41]); foam surface/EC: (*p* < 0.001; 95% CI [−3.91, −1.72])] except for the firm surface/EO condition (*p* > 0.05) (Table 1 and Figure 3). 

Regarding the vision and surface factors, the CoP_vm_ values increased significantly (*p* < 0.001) in the EC condition compared to the EO condition and in the foam surface condition compared to the firm surface condition, no matter what the auditory condition was (Table 1 and Figure 3).

The RI scores were significantly influenced by the auditory factor in both surfaces with EO (firm surface [F = 3.51, T = 3.14, *p* = 0.002, d = 0.87]; foam surface [F = 1.91, T = 3.61, *p* = 0.0007, d = 1.05]) and with EC condition (firm surface [F = 2.29, T= 2.28, *p* = 0.02, d = 0.86]; foam surface [F = 1.23, T = 4.08, *p* = 0.0001, d = 1.13]) with large effect size (d > 0.8) (Table 2). Indeed, listening to preferred music had a positive impact on postural balance. Mainly in the firm surface condition, postural sways decreased by 13.55%. In contrast, listening to non-preferred music negatively affects postural balance and increases postural sways (by 19.13%), mainly in the foam surface condition. Additionally, when assessing the two effects (visual and auditory factors) simultaneously, listening to non-preferred music has a negative impact on postural balance. Indeed, in the EC condition, non-preferred music increased postural sways (20.09% in the firm surface and 23.34% in the foam surface condition) (Table 2).

### 3.2. Dynamic Balance

The three-way ANOVA showed a significant main effect of auditory conditions (F_(2,50)_ = 30.32, *p* < 0.0001, η^2^_p_ = 0.54) on the TUGT values with large effect size (η^2^_p_ > 0.14). The post hoc results showed that the TUGT scores significantly increased when listening to non-preferred music compared to the no music condition (*p* < 0.05; 95% CI [−0.41, −0.17]); whereas when listening to preferred music, the TUGT values significantly decreased compared to both no-music condition (*p* < 0.001; 95% CI [0.2, 0.59]) and the non-preferred music (*p* < 0.001; 95% CI [−0.8, −0.41]) conditions (Table 1 and Figure 3).

## 4. Discussion

This study investigated the effects of listening to preferred vs. non-preferred music on postural performance in middle-aged women and verified if the eventual changes could depend on the music type. As predicted, our findings revealed that listening to certain types of music, particularly preferred music, has effects on the equilibrium system. In particular, significant results were found that listening to self-selected music enhances postural performance in both standing and walking conditions. Previous studies have proved that listening to music improved both static and dynamic balance, and these gains were explained by the possible interactions between the auditory and the equilibrium systems that occur in the peripheral receptors of the inner ear or in the central areas [49,50]. Indeed, the hair cells of the saccule respond to both high-intensity, low-frequency sounds, and vestibular stimuli [22]. Additionally, a stronger connection between the two systems appears in the target region of vestibular inputs and auditory signals (temporo-parietal cortical areas). In fact, various studies have identified a network of cortical and subcortical areas in the parietal and temporal cortex that are multisensory, meaning that they receive vestibular afferents in addition to visual and/or somatosensory inputs [51]. In combination, music also activates the lateral pre-motor and supplementary motor areas [52], which can improve postural performance. 

In accordance with our findings, it has been reported that preferred music benefits postural balance by decreasing the CoP_Vm_ values in adolescents with visual impairment, suggesting that these positive effects of preferred music may be due to its benefits on psychological and physical performances [32,53]. In fact, it has been evidenced that listening to self-selected music led to high positive mood responses [54], perceptions of self-esteem, and feelings of trust and arousal [20]. In addition, Ballmann et al. (2019) found that listening to preferred music enhanced physical performance (as anaerobic performance) by improving feelings of energy and motivation to exercise [55]. In general, it has been suggested that pleasurable musical experiences are related to the dopaminergic reward system, important loci of which include the amygdala, ventral striatum, midbrain, ventral medial prefrontal cortex, and orbitofrontal cortex [31]. However, when the sensory inputs were perturbed, preferred music benefits on static balance were absent. In challenging postural conditions, like maintaining static balance under sensory manipulation (EC and/or foam surface), much cognitive processing is required [56,57,58]. It has been proved that standing quietly requires cognitive processing that increases with increasing the difficulty of the postural task [56,57,59]. During a challenging sensory condition, additional cognitive functioning is needed to manage the cognitive and postural tasks simultaneously. In fact, the reduced peripheral sensory system conditions increased the complexity of the central integrative mechanisms while maintaining postural control and consequently stressed the attentional system [56]. Thus, we assumed that maybe self-selected music gains were not sufficient to induce any cognitive performance’ improvements, which could explain our results. In contrast, our findings showed that preferred music improved dynamic balance by boosting gait performance, suggesting that walking while listening to music may render gait more “automatic” for healthy middle-aged women, which may reduce their risk of falls and enhance their daily life activities. This would, in turn, impact their quality of life, as not only falls but also the fear of falling can affect psychological and social behavior [60]. Future research would benefit from allowing the participants to indicate which music they prefer, as this preference may have a meaningful effect on their postural and gait performances.

On the other side, our results showed that listening to non-preferred music significantly altered both static and dynamic balance in our middle-aged women. Such declines could be due to the fact that when participants listened to music they disliked, they tainted more by paying more attention to the music and less to their physical tasks (such as maintaining balance and walking). Although the effects of preferred music on postural performance have been well explored in different populations, to the best of our knowledge, there is still a paucity of data on non-preferred music effects. However, these negative effects of disliked (non-preferred) music were also observed in mood and cognitive performances [61,62]. Given that both mood and cognitive functions are associated with postural performances, alterations of static and dynamic balance observed in our study following non-preferred music may also be explained by the negative effect of non-preferred music on cognitive and mood functions. These postural alterations could lead to a high risk of falls and injuries, inducing serious problems in their daily lives. Consequently, it is strongly recommended for middle-aged women to avoid listening to disliked music while maintaining balance or walking, which may increase their risk of falls and injuries. 

In the current study, some potential limitations should be addressed in future investigations. First, since our study consisted solely of healthy middle-aged women, generalization of the findings is difficult for other people like women with balance issues or other neurological issues/diagnoses, or young adults or people with a specific disease. Conducting future studies on those populations is highly recommended. Secondly, we suggested that mood and cognition may explain the improvements in postural balance. To confirm this hypothesis, future researchers should explore the effect of preferred versus non-preferred music on mood and cognitive performance. Since music with lyrics influences cognitive functioning [63], it would be very interesting to compare the effects of listening to music with lyrics versus only musical instruments on postural balance. Lastly, as musical grooves are known to influence the neural mechanisms related to balance performance [64], future studies are warranted to explore the effects of different musical grooves.

Since self-selected music seems to offer greater potential for static and dynamic balance performance among middle-aged healthy women, these women are recommended to listen to their preferred music in order to instantly improve their postural performances. In addition, it is strongly recommended that these women avoid listening to disliked or non-preferred music while performing a postural task (like walking or maintaining an upright stand) to prevent them from losing balance or any risk of falls. The outcomes of the present study are likely to have significant practical implications. In fact, postural balance is essential in daily life activities in middle-aged women, as it enhances their functional mobility, coordination, and autonomy [65].

## 5. Conclusions

The outcomes obtained in the current study are clearly in favor of the hypothesis that music influences postural performance in a manner depending on its genre. Indeed, our results showed that preferred music significantly improved both static, mainly during simple tasks, and dynamic balance performance, whereas listening to non-preferred music significantly altered their static balance, mainly during challenging tasks, and their gait capabilities (dynamic balance). Listening to non-preferred music seems to be an attention-demanding activity that negatively affects gait capacities and balance performance in middle-aged women. Interesting practical implications emerge from these results. Healthcare professionals might encourage these women to listen to their preferred music as an effective strategy to promote their balanced performance, improving their quality of life in daily activities. To avoid the risk of falls, middle-aged women are strongly recommended not to listen to their non-preferred music while walking or even during simple postural tasks. This could offer greater potential for everyday functioning and may reduce the risk of falls and dependence, which could improve the quality of life of middle-aged women. 

## Figures and Tables

**Figure 1 healthcare-11-02681-f001:**
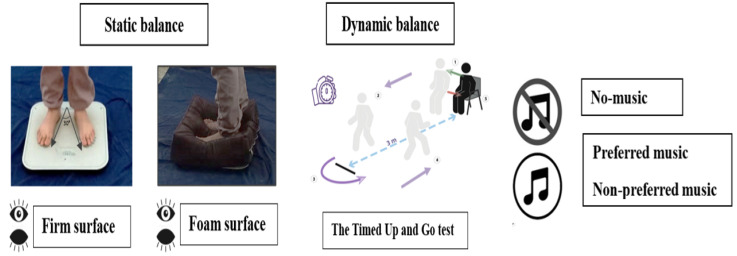
Study design.

**Figure 2 healthcare-11-02681-f002:**
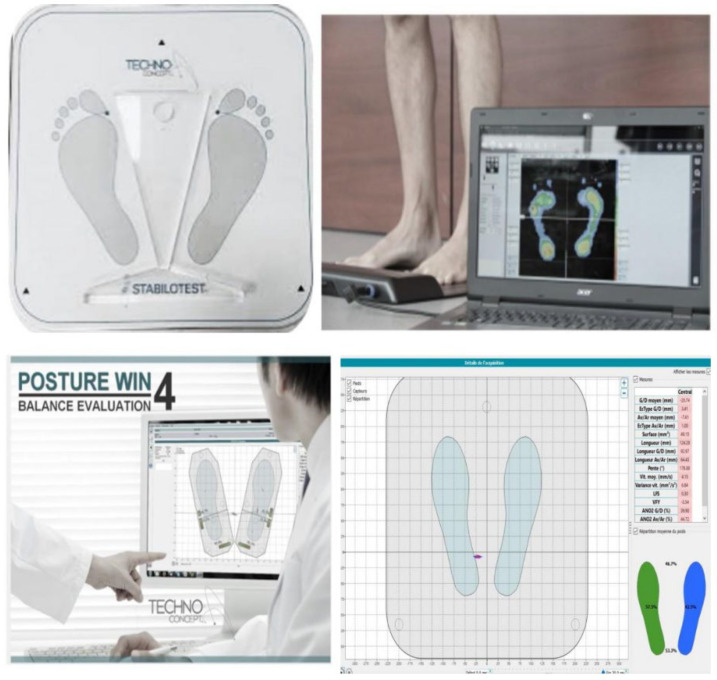
Stabilometric force platform (posture Win© 4).

**Figure 3 healthcare-11-02681-f003:**
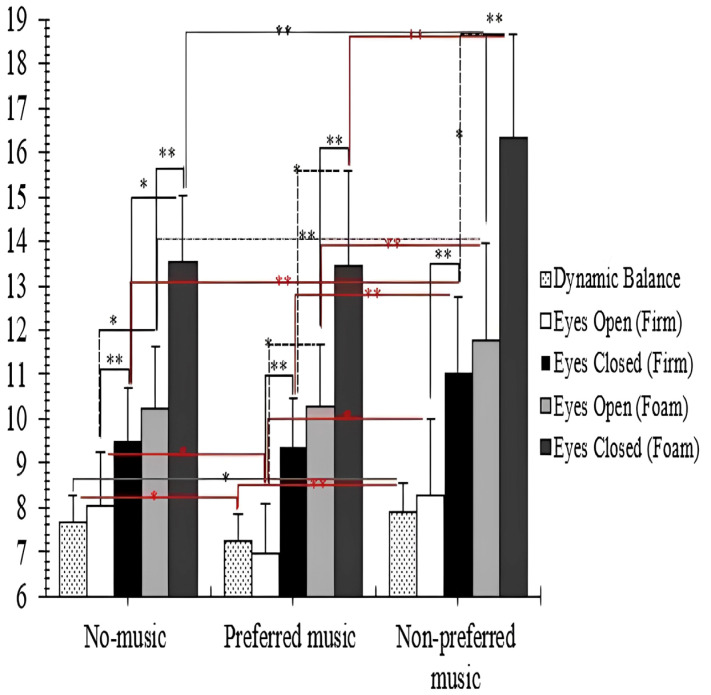
Mean values (with SD) of the static (Center of Pressure Mean Velocity (VmCOP) parameter in both eyes opened (EO) and eyes closed (EC) conditions on the firm and foam surfaces) and dynamic balance during three auditory conditions (preferred music vs. non-preferred music vs. no-music) among middle-aged women. * Significant difference (*p* < 0.05) between no-music vs. Mozart’s Jupiter; ** Significant difference at *p* < 0.01). ― Significant difference between the three auditory conditions (preferred music vs. non-preferred music vs. no-music).

**Table 1 healthcare-11-02681-t001:** Means ± SD of the static (center of pressure mean velocity (CoP_Vm_)) in the eyes open (EO) and eyes closed (EC) conditions on the firm and foam surfaces and dynamic balance (timed up and go test (TUGT)) during the three auditory conditions (preferred music vs. non-preferred music vs. no-music) in middle-aged women.

	No-Music	Preferred Music	Non-Preferred Music
Firm Surface			
EO	8.06 ± 1.24	6.96 ± 0.72 ^+^	8.27 ± 1.28 ^£^
EC	9.49 ± 1.21 ^#^	9.36 ±1.12 ^#^	11.06 ± 1.71 **^#££^
Foam Surface			
EO	10.25 ± 1.4 ^$^	10.28 ± 1.39 ^$^	11.76 ± 2.2 **^#££^
EC	13.52 ± 1.5 ^#$^	13.45 ± 2.15 ^#$^	16.34 ± 2.3 **^#$££^
TUGT (scores)	7.66 ± 0.6	7.26 ± 0.65 ^+^	7.88 ± 0.68 *^££^

^+^ Significant difference (*p* < 0.05) between no-music and preferred music at *p* < 0.05; * Significant difference between no-music and non-preferred music (* at *p* < 0.001 and ** at *p* < 0.001); ^£^ Significant difference between the preferred music and non-preferred music (^£^ at *p* < 0.01 and ^££^ at *p* < 0.001); ^#^ Significant difference (*p* < 0.001) between EO and EC; ^$^ Significant difference (*p* < 0.001) between firm surface and foam surface.

**Table 2 healthcare-11-02681-t002:** Romberg’s Index (RI) scores represent the transition between the different postural conditions under different sensory manipulation: auditory manipulation (preferred music and non-preferred music) in the eyes open (EO) (RI1) and eyes closed (EC) (RI2) conditions on the firm and foam surfaces in middle-aged women.

	RI1-Auditory Contribution (EOA/EO)		RI2-Auditory Contribution with EC (ECA/EC)	
	*Preferred* *Music*	*Non-Preferred* *Music*	*Preferred* *Music*	*Non-Preferred* *Music*
**Firm Surface**	−13.55% (0.15)	6.64% (0.28) **	−0.07% (0.16)	20.09% (0.23) *
**Foam Surface**	2.33% (0.19)	19.13% (0.29) ***	0.66% (0.21)	23.34% (0.18) ***

* Significant difference between the RI of preferred music vs. non-preferred music at *p* < 0.05, ** at *p* < 0.01, *** at *p* < 0.001.

## Data Availability

The datasets used and/or analyzed during the current study are available from the corresponding author upon reasonable request.

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
