# Peer review of "The Influence of Listening to Preferred versus Non-Preferred Music on Static and Dynamic Balance in Middle-Aged Women"

_healthcare, 2023, doi:10.3390/healthcare11192681_

Round 1

Reviewer 1 Report

This is an interesting paper with moderate relevance to the readership. It revolves around a topic that is not commonly addressed in music perception studies. As such it opens new perspectives for future research, which is applaudable. The paper is well written with good readability and understandability, and the methodology seems to be sound. The language use is mostly OK, though there are some minor grammatical mistakes. The theoretical background, however, is rather limited and there is a danger of generalizing too much from too little data in the conclusions section. I would not be opposed to accept the paper for publication, on condition that some concerns are addressed appropriately. I list some of them below. The results, however, are not quite substantial. They could perhaps have a better place in a broader study where they could function as a kind of pilot study.

General remarks

·      The language use is relatively fluent, but there are some grammatical errors (e.g. verbs; plural, singular)

·      The theoretical background could be more elaborate. This holds for the empirical background on postural equilibrium and the relationship between the auditory and vestibular system.

·      The methodology is clearly explained and is quite understandable.

·      The significances in table 1 are difficult to read. A more transparent visualization should make things easier.

·      The description of the independent variable “music” is rather loose. There are so many kinds of music and musical genres, which all can have very different effects listeners, which also have different personalities and learning histories. Much more could be done by inserting more independent variables in the analysis, to make the conclusions more fine-tuned. The paper therefore provides only some hints to a possible more elaborate study but the empirical findings are not very strong.

Detailed comments

·      Page 2, 51: “falls lead to morbidity”. This seems not obvious. One would expect that falls lead to injuries or traumata but not to sickness, which may have another etiology. Please explain a little more to motivate this claim.

·      Page 2, line 53: “central nervous system” instead of “center nervous system”

·      Page 2, lines 56 ff: “music leads to …”. This is all very general. A more detailed description and motivation is needed here.

·      Page 2, line 63; listening to music. Which kind of music? Using such a general category reduces considerably its informative value.

·      Page 2, line 80: this sentence seems to be ill-formulated; what is the meaning of “its effects steel scare”?

·      Page 3, line 106: grammatical error: “all women given”. Should this be “all women gave”?

·      Page 4, line 139: “platform” instead of “plateform”?

·      Page 4, line 167: TUGT: provide full description of the abbreviation at first appearance of the term.

·      Page 5, line 170: asked to “sit” instead of “sat”?

·      Page 5, line 176: same remark: “sit”

·      Page 5, line 177: Close quotation marks after Go. “Go”.

·      Page 5, line 180: “participants “were” sitting” instead of “was” sitting

·      Page 5, lines 186 and 201: there is some confusion about two-way of three-way ANOVA. Please explain clearly.

·      Page 6, table 1. The significant differences between the independent variables is not easy to read. Perhaps another way of visualization can be more transparent (histogram with asterisks above the bridges between the columns?).

·      Page 7, line 266: is there no contradiction in speaking of “peripheral” receptors of the “inner” ear? Please motivate a little more, as peripheral receptors are mostly located in the skin. The ear drum could be considered to be peripheral, but the inner ear? There is, of course, a distinction between the auditory nerve which transduces information from the periphery to more central areas of the brain. Some more in-depth discussion of this terminological use is needed here.

·      Page 7, line 282: this is a dangerous claim. Do challenging conditions require more “cognitive” processing? In case of danger, e.g., more innate reflex-like behavior can be evoked as well. There is also only one reference here. A stronger motivation is needed to make this claim plausible.

·      Page 7, line 300: delete the comma after “given that,”

·      Page 8, line 11: this sentence seems to be awkwardly constructed. Pease reword.

The overall style of writing is rather fluent, but there are many grammatical mistakes which should be corrected. Carefull check of the langage use is needed.

Author Response

Dear Reviewer

We thank you for studying the article and the recommendations made.

They helped us.

Reviewer 2 Report

Thank you for this unique and interesting manuscript. I can see a lot of potential for use with patients. 

Please review for the English language, punctuation (commas, end brackets, etc.), tenses (majority should be in past tense), type-o's, and wording (missing words, confusing wording, etc.).

Some specific questions, comments, and/or changes are listed here:

1. Page 2, line 57 - "boost" should be "boosts"

2. Page 2, line 62 - "listing" should be "listening"

3. Page 2, line 74 - "proved" may be too strong of a word. Perhaps you might consider "demonstrated"?

4. Page 2, lines 79-80 - this sentence is confusing.

5. Page 3, line 106 - "given" should be "gave"

6. Page 3-4 -  these illustrations were nice and helpful

7. Page 4, line 148 - "trails" should be "trials"

8. Page 5, line 170 and line 176 - "sat" should be "sit"

9. For future studies, do you recommend including women with balance issues or other neurological issues/diagnoses?

10. How was the music chosen? Did you provide a list of genres or songs? 

11. How did participants pick out their preferred music? How did they pick out their non-preferred music?

12. Did you have access to a wide variety of music or were there only limited selections to choose from?

13. Did the music have words or was it only instrumental? This could have an impact on cognitive functioning as well.

14. Would  you please clarify - did all women participate in all 3 conditions? How much time was there between the sessions? Was the first session the session with no music? 

15. Were the conditions presented in the same order for all participants? 

16. What types of genres were used?

17. Why was the sample size so small?

18. Please be consistent in your references - for example, capitalizing the names of journals

This could be improved upon. Some of the wording is confusing, in some cases there are missing words, and in some cases words are used in an unfamiliar manner. 

Author Response

(The authors gave the same response as above.)
